# Long Diagnostic Delay with Unknown Transmission Route Inversely Correlates with the Subsequent Doubling Time of Coronavirus Disease 2019 in Japan, February–March 2020

**DOI:** 10.3390/ijerph18073377

**Published:** 2021-03-24

**Authors:** Tsuyoshi Ogata, Hideo Tanaka

**Affiliations:** 1Tsuchiura Public Health Center of Ibaraki Prefectural Government, Tsuchiura 300-0812, Japan; 2Fujiidera Public Health Center of Osaka Prefectural Government, Fujiidera 583-0024, Japan; TanakaH61@mbox.pref.osaka.lg.jp

**Keywords:** COVID-19, diagnostic delay, doubling time, unknown exposure, Japan

## Abstract

Long diagnostic delays (LDDs) may decrease the effectiveness of patient isolation in reducing subsequent transmission of coronavirus disease 2019 (COVID-19). This study aims to investigate the correlation between the proportion of LDD of COVID-19 patients with unknown transmission routes and the subsequent doubling time. LDD was defined as the duration between COVID-19 symptom onset and confirmation ≥6 days. We investigated the geographic correlation between the LDD proportion among 369 confirmed COVID-19 patients with symptom onset between the 9th and 11th week and the subsequent doubling time for 717 patients in the 12th–13th week among the six prefectures. The doubling time on March 29 (the end of the 13th week) ranged from 4.67 days in Chiba to 22.2 days in Aichi. Using a Pearson’s product-moment correlation (*p*-value = 0.00182) and multiple regression analyses that were adjusted for sex and age (correlation coefficient −0.729, 95% confidence interval: −0.923–−0.535, *p*-value = 0.0179), the proportion of LDD for unknown exposure patients was correlated inversely with the base 10 logarithm of the subsequent doubling time. The LDD for unknown exposure patients was correlated significantly and inversely with the subsequent doubling time.

## 1. Introduction

An outbreak of novel coronavirus disease 2019 (COVID-19) began in December 2019 in China. It spread to other countries, including Japan, and the World Health Organization declared it a public health emergency of international concern [1].

In Japan, in the first wave, public health centers implemented contact tracing under the direction of the “Cluster response task force” of the National Government, and contacts of known exposure patients promptly underwent polymerase chain reaction (PCR) tests. On the other hand, for mildly affected patients without known exposure, the National Government determined that they consult a public health center or visit a physician only after waiting four days from the time of symptom onset [2,3,4,5,6]. The “criterion of waiting for four days” could increase the proportion of long diagnostic delay (LDD), and subsequently, the number of local patients. We reported in a previous study that the LDD for patients with unknown exposure was 65%, significantly higher than the LDD for known exposure patients (adjusted odds ratio: 2.38, 95% confidence interval [CI]: 1.354–4.21) [7].

LDDs ≥6 days may decrease the effectiveness of interventions that involve the isolation of patients to reduce the risk of subsequent transmissions [8,9]. Since the close contacts of patients or patient clusters are under observation and directed to stay at home, their diagnostic delay may not influence the increase in the number of patients. However, LDDs of patients with unknown exposure may influence a local increase in patient numbers. No studies have reported the association between LDD in patients with known or unknown exposures and any subsequent increase in COVID-19 incidence.

We hypothesized that a LDD, especially for patients with unknown exposure, may influence the increase in the number of COVID-19 patients. Therefore, this study aimed to elucidate the correlation of the proportion of LDD with a subsequent doubling time, the index for the increase in the number of COVID-19 patients.

## 2. Materials and Methods

### 2.1. Study Design

The study used an ecological, epidemiological design. We conducted geographic correlations.

### 2.2. Study Setting and Participants

Japan consists of 47 prefectures. A prefecture is a governmental body and a subdivision. In this study, we included those prefectures with >30 reported COVID-19 patients as of 22 March, 2020 (the end of the 12th week). Therefore, eight prefectures, i.e., Hokkaido, Saitama, Chiba, Tokyo, Kanagawa, Aichi, Osaka, and Hyogo, were eligible. However, we could not retrieve the data on the route of exposure of patients in Tokyo. We excluded Hokkaido because the Governor of that prefecture, responding to the increased number of COVID-19 patients, declared a situation of emergency and requested that people stay at home on 28 February (the 9th week), 2020 [10]. Finally, six prefectures, i.e., Saitama, with a population of 7.3 million people, Chiba, with 6.3 million, Kanagawa, with 9.2 million, Aichi, with 7.6 million, Osaka, with 8.8 million, and Hyogo, with 5.5 million, were eligible.

Although in two out of the six prefectures, the number of patients with symptom onset at the end of the eighth week was less than five, the numbers of patients with symptom onset at the end of the ninth week in all six prefectures were not less than five. Therefore, we selected symptomatic COVID-19 patients with a date of onset in and after the ninth week (beginning on 24 February 2020) in each prefecture.

### 2.3. LDD

We defined diagnostic delay in this study as the duration between the date of symptom onset and the date of confirmation of severe acute respiratory syndrome coronavirus 2 (SARS-CoV-2) positivity by PCR test. A patient was considered to have a LDD if the duration between COVID-19 symptom onset and confirmation was ≥6 days, because such a delay could reduce the intervention (patient isolation) effect [9].

To calculate the LDD, we included COVID-19 patients from the six prefectures whose onset of symptoms was dated between 24 February and 15 March, 2020, (the 9th and 11th weeks), and SARS-CoV-2 positive patients as confirmed by a PCR test. We excluded patients who were asymptomatic and those whose symptom onset dates were missing.

We used publicly available anonymized data from 22 local government official websites in the six prefectures (Table A1). The data retrieved included sex, age categories, date of onset of the subjective symptoms, date of confirmation of SARS-CoV-2 positivity by PCR tests, and exposure routes (route of patient detection). The routes of exposure were classified according to the information disclosed at the time of diagnosis as “known” (having had contact with an infected patient or visited a place of an outbreak), “imported” (from foreign countries during the incubation period), or “unknown.” We censored the data after 5 April, 2020 (the end of the 14th week) because data in two prefectures could not be retrieved in the 15th week. The number of patients with known exposures, unknown exposure, and those who were imported, were 207, 124, and 38, respectively [7].

For the 207 patients with known and 124 patients with unknown exposures, the proportion of patients who met the LDD definition (≥6 days) in each prefecture was calculated for patients with the date of symptom onset between 24 February and 15 March (the 9th and 11th week).

### 2.4. Doubling Time

We calculated the doubling time by counting the number of patients with the date of symptom onset on 24 February (the first day of the ninth week) in each prefecture. The doubling time was estimated by dividing the natural logarithm of two by the growth rate [11,12]. We could not obtain the number of patients with onset in the 14th week because detailed data on COVID-19 patients were not completely disclosed in the 15th week in two prefectures because of the rapid increase in the number of COVID-19 patients. Therefore, we adopted a 14-day moving-average as the doubling time for patients with symptom onset between 15 March and 29 March (the end of the 11th and 13th week).

In addition to patients with symptom onset between 24 February and 15 March (from the 9th to the 11th week), we retrieved publicly available anonymized data of patients with symptom onset between 16 March and 29 March (the 12th and the 13th week) in the six prefectures, from the data provided by the 22 local government official websites in the six prefectures (Table A1).

### 2.5. Statistical Analysis

We calculated the diagnostic delay in COVID-19 patients with symptom onset between 24 February and 15 March (from the 9th to the 11th week) and we estimated the change in moving-average doubling time for 14 days among the six prefectures.

We calculated the correlation (*p*-value) between the explanatory factor during the 9th to the 11th week and base 10 logarithm of the moving-average doubling time for 14 days in the 12th and 13th week in the prefectures using a Pearson’s product-moment correlation. The explanatory variable included the proportion of LDD for patients with unknown exposure and onset in the 9th–11th week, proportion of LDD for patients with known exposure and onset in the 9th–11th week, sex, and age. Using multiple regression analyses, we calculated the correlation (correlation coefficient, 95% CI, and *p*-value) between the proportion of LDD for patients with unknown exposure and onset in the 9th–11th week and base 10 logarithm of the moving-average doubling time for 14 days in the 12th and 13th week in the prefectures. We also separately calculated the correlation between the proportion of LDD for patients with unknown exposure and base 10 logarithm of the moving-average doubling time.

We implemented similar analyses using ≥5 days and ≥7 days instead of ≥6 days as the definition of LDD. Statistical analyses were performed using R (version 3.6-2; The R Foundation for Statistical Computing, Vienna, Austria).

### 2.6. Ethical Concerns

The study used only publicly available anonymized data. The study did not use patients’ data from other sources. The study was approved by the Tsuchiura Public Health Center of Ibaraki Prefectural Government (protocol number: Tsuchi-Ho R20–01).

## 3. Results

We included 369 patients with an onset of symptoms between 24 February and 15 March (from the 9th to the 11th week) in the study. The proportion of symptomatic patients for whom we could not acquire data on the date of onset of COVID-19 reported between 2 March and 22 March (the 10th and 12th week) in each prefecture ranged from 0% in Saitama Prefecture to 8.5% in Osaka Prefecture. Data on the onset of symptoms or diagnostic delay were obtained for most COVID-19 patients during the period in each prefecture. Table 1 shows the demographic data of patients with symptom onset between 24 February and 15 March (from the 9th to the 11th week), LDD and the mean diagnostic delay by each prefecture.

In addition to the 369 patients, we retrieved data on 717 patients with the onset of symptoms between 16 March and 29 March (the 12th and 13th week), in the six prefectures. Figure 1 shows the 14 day moving-average of the doubling time of COVID-19 derived by counting the number of patients with an onset of symptoms between 24 February (the first day of the 9th week) and 29 March (the last day of the 13th week). The doubling time in Aichi and Hyogo increased during this period. It decreased from mid-March in Osaka, Saitama, Kanagawa, and Chiba. The doubling time on 29 March (the end of the 13th week) was 22.2 days in Aichi, 12.8 days in Hyogo, 8.47 days in Osaka, 8.11 days in Saitama, 6.75 days in Kanagawa, and 4.67 days in Chiba. 

The scatter plots of the six prefectures with the proportions of LDD for the 122 patients with unknown exposure and the doubling time in the 12th–13th week are shown in Figure 2. Figure 1 shows trajectory of the doubling time as time passing, while Figure 2 shows its geographic correlation with the proportion of long diagnostic delay.

The proportion of LDD for patients with unknown exposure and onset in the 9th–11th week was correlated with the base 10 logarithm of the doubling time in the prefectures using a Pearson’s product-moment correlation test (*p*-value = 0.00182). After adjusting for sex and age, the analyses revealed that the proportion of LDD for unknown exposure patients was correlated inversely with the base 10 logarithm of the subsequent doubling time (correlation coefficient −0.729, 95% CI: −0.923–−0.535, *p*-value = 0.0179).

The proportion of LDD for patients with known exposure and onset in the 9th–11th week was not correlated with the base 10 logarithm of the doubling time among the prefectures by Pearson’s product-moment correlation test (*p*-value = 0.2171). After adjusting sex and age, the proportion of LDD for patients with known exposure was not correlated with the base 10 logarithm of the subsequent doubling time (correlation coefficient −0.691, 95% CI: −1.927–0.544, *p*-value = 0.387). (Table 2, Figure 2)

When we changed the definition of LDD, a similar inverse correlation was observed for patients with unknown exposure for the ≥5 days definition (Pearson’s product-moment correlation *p*-value = 0.021; multiple regression analysis correlation coefficient −1.955, 95% CI: −2.700–−1.211, *p*-value = 0.0357) and for the ≥7 days definition (Pearson’s product-moment correlation *p* = 0.001; multiple regression analysis correlation coefficient −0.758, 95% CI: −0.929–−0.588, *p*-value = 0.0129),

## 4. Discussion

The doubling time varied among the prefectures. The proportion of LDD for unknown exposure patients was correlated inversely with the logarithm of the subsequent doubling time. We obtained similar results when we changed the definition of LDD to ≥5 days and ≥7 days instead of ≥6 days.

Mild to moderately affected COVID-19 patients transmitted the virus to contacts or shed the live virus within five to eight days after symptom onset [11,12,13]. The secondary attack rate was low among those initially exposed to infected patients after six days of the onset of symptoms [9]. Therefore, the isolation of patients with LDD ≥6 days would have limited the preventive effects on subsequent viral transmission. LDDs may decrease the effectiveness of interventions involving patients’ isolation in order to reduce the risk of subsequent transmissions.

A study using a mathematical model showed that the delay between symptom onset and isolation played a major role in controlling the COVID-19 outbreak, and that the probability of achieving outbreak control dropped when the isolation delay increased by 8.1 days, compared to a short delay of 3.4 days [8]. From that study, a positive correlation between long diagnostic delay and a greater increase in the number of subsequent COVID-19 patients was expected. These findings were supported by the results of the present study; a shorter doubling time indicated a greater increase in the number of subsequent COVID-19 patients.

The mean incubation period of COVID-19 is approximately five days [14]. Since COVID-19 patients transmit the virus to contacts from two days prior to the onset of symptoms and the median serial interval of COVID-19 was reported as four to five days [15,16], the secondary patients may not be prevented by the containment of the index patient. Nevertheless, containment is thought to effectively prevent virus spread to third-generation patients using a staying at home strategy for second-generation presymptomatic patients through contact tracing (Figure 3a). If a patient with unknown exposure has LDD, the secondary patient’s restriction of behavior is delayed, and transmission from the secondary patient will be prolonged (Figure 3b). Thus, the geographic area with a higher proportion of LDD would have a higher risk of disseminating to third-generation patients, with a subsequent increase in the reproductive number of COVID-19 infection in the area.

Although the proportion of LDD for unknown exposure patients was correlated inversely with the subsequent doubling time, the proportion of LDD with known exposure was not correlated significantly with the subsequent doubling time. Since responding to a patient or a cluster of patients leads to contact tracing, and the close contacts may be directed to stay at home by the public health center, the behavior of a patient with known exposure would generally have been restricted by the time the specimen was collected (Figure 3a: Case A2). Therefore, the infectivity of patients with known exposure is estimated to be low, regardless of the period of diagnostic delay of the patient, and does not significantly influence the subsequent spread of the virus.

We could not find previous studies on the association between diagnostic delays for COVID-19 patients with known or unknown exposures and increase in the number of patients. The fact that a large proportion of LDD for patients with unknown exposures may bring about a subsequent increase in transmission may be useful in the development of intervention strategies against COVID-19. For the prevention of further spread of transmission, PCR tests should be conducted earlier, i.e., from symptom onset in patients, even if they do not have any epidemiological link with COVID-19. However, given the low level of PCR testing in Japan, these results cannot be validated externally for countries with a higher level of PCR testing.

This study had several limitations. First, this study used cross-sectional and ecological designs; thus, the results did not prove any causal relationships. Moreover, ecological studies can cause fallacy. Second, the data were collected only through websites and their authenticity could not be confirmed. Third, differences in the management and disclosure of relevant data may exist among the prefectures that influenced the results. Fourth, due to the limited number of PCR tests, the data did not reflect all the patients. Finally, we could not collect relevant data after the 14th week.

Further investigations, both in Japan and other countries are necessary to adequately assess the impact of LDD on the subsequent increase in the number of COVID-19 based on the exposure route. Investigations on associations between the latent time for patients with known exposures and the secondary attack rates for contacts will also be meaningful.

## 5. Conclusions

The proportion of LDD among patients with unknown exposure was correlated inversely with the base 10 logarithm of the subsequent doubling time of the incidence, indicating a substantial increase on the regional spread of this virus in Japan. The data and outcomes of this study may be useful in the development of intervention strategies against COVID-19. Further investigations are necessary to assess impact of LDD on the subsequent increase in the number of COVID-19 based on the exposure route.

## Figures and Tables

**Figure 1 ijerph-18-03377-f001:**
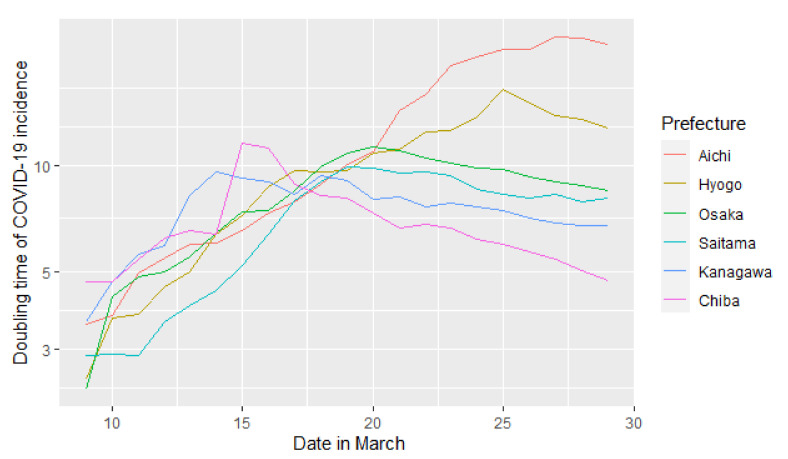
The 14 day moving-average of doubling time derived by counting the number of patients with the date of symptom onset beginning on 24 February (the first day of the 9th week) in each prefecture. The longitudinal axis is expressed as the natural logarithm.

**Figure 2 ijerph-18-03377-f002:**
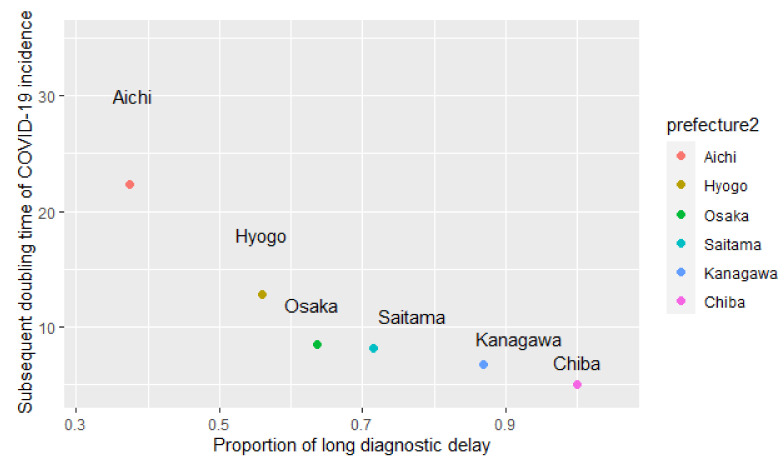
Geographic correlation between the proportion of long diagnostic delay for patients with unknown exposure (the 9th to the 11th week, 2020) and subsequent 14 days moving-average doubling time of COVID-19 incidence (the 12th–13th week) among the six Japanese prefectures.

**Figure 3 ijerph-18-03377-f003:**
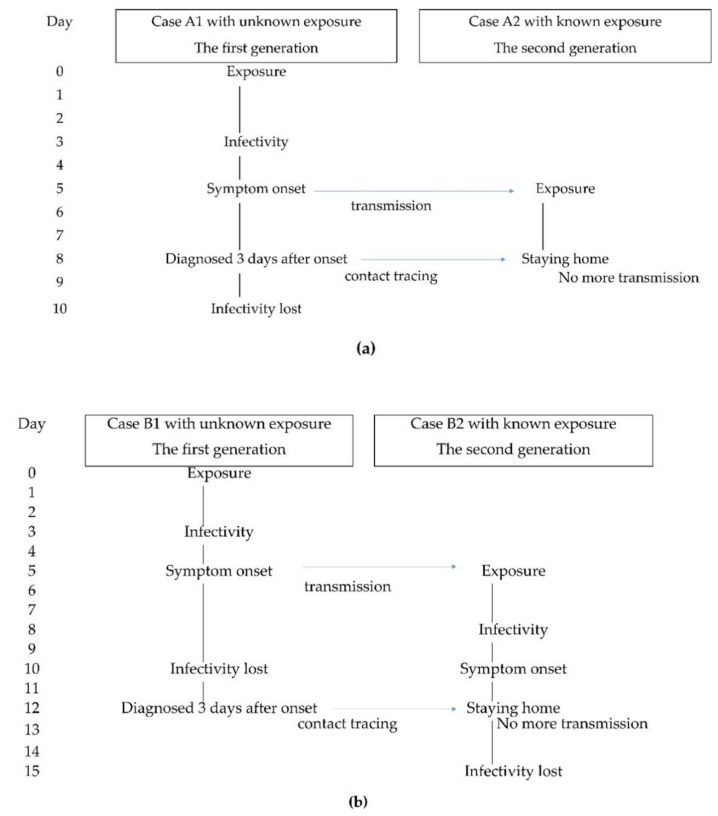
An example of the transmission links in COVID-19, assuming a patient with a symptom onset of five days after exposure and infectivity of between 3 to 10 days after exposure; if the first generation of the link transmits the virus to the second generation on day five. (**a**) In this example, Case A1, the first generation, did not have a known exposure, but was diagnosed and isolated three days after onset on day eight. The isolation of Case A1 cannot prevent the transmission to Case A2, the second generation. However, Case A2 was directed to stay at home early on based on the contact tracing of Case A1, regardless of the period of reporting the delay of Case A2. The staying at home of Case A2 limited the transmission from Case A2. (**b**) In this example, Case B1, the first generation, did not have a known exposure, but was diagnosed and isolated on day 12, seven days after the onset. Case B2, the second generation, is directed to stay at home late, based on the contact tracing of Case B1. Transmission from Case B2 continued until day 12.

**Table 1 ijerph-18-03377-t001:** Demographic data and diagnostic delay in COVID-19 patients with symptom onset between 24 February and 15 March (from the 9th to the 11th week) by prefecture.

Prefecture	Total	Male Sex	Age ≥ 60	Patients with Long Diagnostic Delay	Mean Diagnostic Delay
	*N*	*N (%)*	*N (%)*	*N (%)*	Days (SD *)
Aichi	102	47 (46)	68 (67)	18 (18)	2.9 (3.7)
Hyogo	73	36 (49)	40 (55)	45 (62)	7.7 (4.5)
Osaka	89	44 (49)	25 (28)	64 (72)	8.3 (4.6)
Saitama	39	20 (51)	11 (28)	24 (62)	7.8 (3.7)
Kanagawa	43	26 (60)	21 (49)	33 (77)	8.0 (2.5)
Chiba	23	13 (57)	13 (57)	17 (74)	9.0 (4.6)

* SD: Standard deviation.

**Table 2 ijerph-18-03377-t002:** Geographic correlation between the proportion of long diagnostic delay (≥6 days) in symptomatic COVID-19. Patients (the 9th–11th week) and the base 10 logarithm of the doubling time of the COVID-19 incidence (the 12th–13th week) among the six Japanese prefectures using a Pearson’s product-moment correlation and multivariate regression analyses.

Variables	Pearson’s Product-Moment Correlation	Multiple Regression Analysis 1 *		Multiple Regression Analysis 2 *	
	*p*-Value	*p*-Value	Correlation Coefficients (95% Confidence Interval)	*p*-Value	Correlation Coefficients (95% Confidence Interval)
Proportion of LDD for patients with unknown exposure and onset in the 9th–11th week	0.00182	0.0179	−0.729 (−0.923–−0.535)		
Proportion of LDD for patients with known exposure and onset in the 9th–11th week	0.2171			0.387	−0.691 (−1.927–0.544)
Sex	0.055	0.256	−0.180 (−0.404–−0.044)	0.603	0.828 (−1.826–−3.482)
Age ≥ 60	0.400	0.062	0.732 (0.358–1.106)	0.949	−0.045 (−1.267–1.176)

* Two multiple regression analyses were implemented for patients with unknown or known exposure separately.

## Data Availability

The data is publicly available at Figshare repository; doi10.6084/m9.figshare.14253302.

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
