# Peer review of "Long Diagnostic Delay with Unknown Transmission Route Inversely Correlates with the Subsequent Doubling Time of Coronavirus Disease 2019 in Japan, February–March 2020"

_ijerph, 2021, doi:10.3390/ijerph18073377_

Round 1

Reviewer 1 Report

Review of: ”Long Diagnostic Delay with Unknown Transmission Route 2 and the Subsequent Doubling Time of Coronavirus Disease 3 2019 in Japan”.

This is a study aims at investigating the correlation between the proportion of patients with Long diagnostic delays LDD of COVID-19 cases with unknown transmission routes and the subsequent doubling time through an ecological study design and regression. From mathematical modelling a positive correlation would be expected, however they identify a negative association.

The authors include 6 different prefectures in Japan, from which they can access data from different webpages, and have access to a sample of 118 cases with known and 209 case with unknown exposures. The prefectures are all relatively large, the smallest has 5,2 million habitants. The sample representing all the prefectures is relatively small, but the study is also from the start of the pandemic. It is not clear for me whether the sample consists of all identified persons tested positive in the inclusions period all different subsamples?  Also, it is not clear how many persons the proportion of LLD builds on in the different prefectures please specify this in the paper.

Long Diagnostic Delay (LDD) is defined proportion of persons with more than 6 days from symptoms to positive test. Expecting that most persons has isolated them self after symptoms has started, it is not so strange that they do not find a positive association. To include the asymptotic period would be very interesting. Is this possible for the persons with known exposure?

Also, instead of using proportion of persons with more that 6 days from symptoms to positive test, average time and SD from the different regions does not through away so much information.

The authors includes  the following 3 explanatory variables: the proportion of LDD for cases with unknown exposure and onset in the 9th-11th week, proportion of LDD for cases with known exposure and onset in the 9th-11th week, proportion of cases in the 10th-11th week out of cases in the 9th-11th week, and number of tests per one million of the population in the 11th week. The first are the variables of interest and the last (tests per one million makes sence). However, “proportion of cases in the 10th-11th week out of cases in the 9th-11th week” seems like adjusting for the outcome. This study should be seen as a cross sectional study and not repeated measures, I will therefore not commend to adjust for the variable.

Also, the description of statistics says that only variables with a p-value less than 0,2 are included, several studies has shown that this is not a good strategy. As we only measure 3 explanatory variables, and one of them is just suggested to be deleted I would be suggest to make a unadjusted model and a adjusted model.

In table 1 an Analysis 1 and an Analysis 2 is presented, please make clear which variables that are included in which analysis. I expect that known exposures and unknown exposures are analyzed separately?

Age and sex are mentioned as measured variables, but these are not mentioned in the result section, please show results for these for each region together with N and distribution of  LDD.   

Reviewer 2 Report

Overall assessment

This report is very important in management of COVID-19 infection which describes the use of mathematical model to show the importance of delay between symptom onset and the confirmation of infection, this had a major role in the  control of COVID-19 outbreak. The results of the study showed that there is a probability of achieving outbreak control when the isolation delay increased by 8.1 days.

Minor Revisions

  1. Line 56: study Design: the authors used “ prefectures” probably to indicate Directorates of regions in the country, I suggest to use Directorates if possible …but consultation with linguistics may be useful.
  2. Figures 1 and 2: the two figures are almost show the same idea, therefore I recommend to include Figure 1 and omitting Figure 2.
  3. Did the authors selected or excluded cases from certain Directorates “prefectures” , this may be clarified and described the exclusion criteria

Reviewer 3 Report

Dear authors, thank you for this correlation study. The paper needs some minor corrections:

lines 86-88: you gave inverse numbers respect to known and unknown different cases

line 98: please correct January 2121

line 158-159 & others: 11th week, please correct 11st, as well as the 12-13th week.

The authors highlight that the proportion of LDD for unknown exposure patients inversely correlated with the subsequent doubling time of COVID-19 cases, that is quite intuitive/obvious. Besides, in lines 239-240 the authors state that the outcomes of this study can be useful in the development of intervention strategies against COVD-19. Anyway, which intervention strategy could be applied is not clear, considering the unknown exposure. Although the authors well conducted this study, there are a number of limits - listed by the authors as well - that reduce the utility of the results. For these reasons, I would suggest to submit the manuscript to another journal. Further, I would improve the title, i.e. "Long Diagnostic Delay with Unknown Transmission Route inversely correlates with the Subsequent Doubling Time of COVID-19 cases, Japan, February-March 2020".

Round 2

Reviewer 1 Report

I have no further comments 

Reviewer 3 Report

The authors have well answered all the comments from the reviewers, and improved data analysis accordingly, thank you.